# Aligning Synthetic Medical Images with Clinical Knowledge using Human Feedback

**Shenghuan Sun** *
University of California, San Francisco
shenghuan.sun@ucsf.edu

**Gregory M. Goldgof** *
Memorial Sloan Kettering Cancer Center
goldgofg@mskcc.org

**Atul Butte**
University of California, San Francisco
atul.butte@ucsf.edu

**Ahmed M. Alaa**
UC Berkeley and UCSF
amalaa@berkeley.edu

## Abstract

Generative models capable of capturing nuanced clinical features in medical images hold great promise for facilitating clinical data sharing, enhancing rare disease datasets, and efficiently synthesizing annotated medical images at scale. Despite their potential, assessing the quality of synthetic medical images remains a challenge. While modern generative models can synthesize visually-realistic medical images, the clinical validity of these images may be called into question. Domain-agnostic scores, such as FID score, precision, and recall, cannot incorporate clinical knowledge and are, therefore, not suitable for assessing clinical sensibility. Additionally, there are numerous unpredictable ways in which generative models may fail to synthesize clinically plausible images, making it challenging to anticipate potential failures and manually design scores for their detection. To address these challenges, this paper introduces a *pathologist-in-the-loop* framework for generating clinically-plausible synthetic medical images. Starting with a diffusion model pretrained using real images, our framework comprises three steps: (1) evaluating the generated images by expert pathologists to assess whether they satisfy clinical desiderata, (2) training a reward model that predicts the pathologist feedback on new samples, and (3) incorporating expert knowledge into the diffusion model by using the reward model to inform a finetuning objective. We show that human feedback significantly improves the quality of synthetic images in terms of fidelity, diversity, utility in downstream applications, and plausibility as evaluated by experts. We also show that human feedback can teach the model new clinical concepts not annotated in the original training data. Our results demonstrate the value of incorporating human feedback in clinical applications where generative models may struggle to capture extensive domain knowledge from raw data alone.

## 1 Introduction

Diffusion models have recently shown incredible success in the conditional generation of high-fideltiy natural, stylized and artistic images [1–6]. The generative capabilities of these models can be leveraged to create synthetic data in application domains where obtaining large-scale annotated datasets is challenging. The medical imaging field is one such domain, where there is often a difficulty in obtaining high-quality labeled datasets [7]. This difficulty may stem from the regulatory hurdles that impede data sharing [8], the costs involved in getting experts to manually annotate images [9], or the natural scarcity of data in rare diseases [10]. Generative (diffusion) models may provide a partial solution to these problems by synthesizing high-fidelity medical images that can be easily shared among researchers to replace or augment real data in downstream modeling applications [11, 12].

37th Conference on Neural Information Processing Systems (NeurIPS 2023).

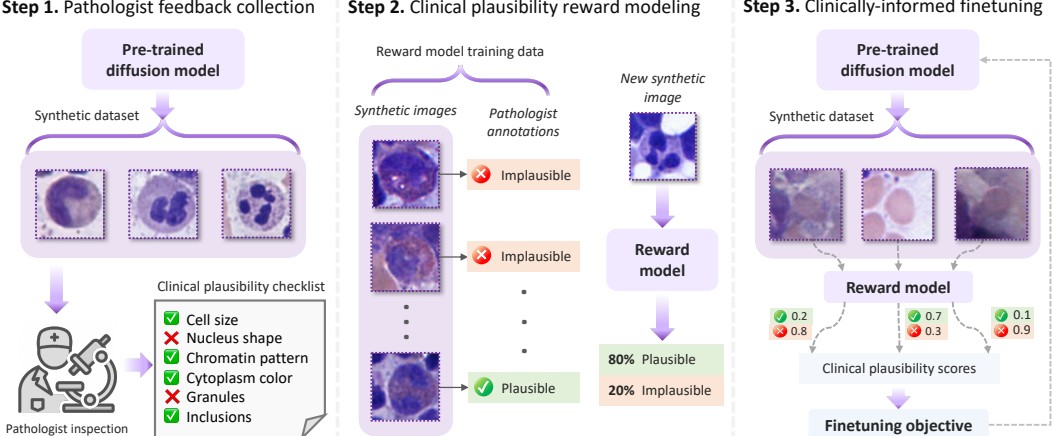

Figure 1: **Overview of our pathologist-in-the-loop synthetic data generation framework. (1)** In Step 1, a synthetic dataset is sampled from a generative model pretrained using a dataset of real medical images. The dataset is then inspected by a pathologist who examines each image to determine its plausibility based on a set of criteria. For each image, the pathologist provides binary feedback, labeling a synthetic image as "1" if it fails to meet all of the plausibility criteria. **(2)** In Step 2, the synthetic images and pathologist feedback obtained in Step 1 are used to train a reward model that predicts human feedback on new images. **(3)** Finally, the generative model is finetuned via an objective function that uses the reward model to incentivize the generation of clinically plausible images.

**What sets medical image synthesis apart from image generation in other fields?** Unlike mainstream generative modeling applications that prioritize visually realistic or artistically expressive images, synthetic medical images require a different approach. They must be grounded in objective clinical and biological knowledge, and as such, they leave no room for creative or unrestricted generation. Given that the ultimate goal of synthetic medical images is to be used in downstream modeling and analysis, they must faithfully reflect nuanced features that represent various clinical concepts, such as cell types [13], disease subtypes [7], and anatomies [14]. Off-the-shelf image generation models are not capable of recognizing or generating clinical concepts, rendering them unsuitable for generating plausible medical images without further adaptation [15, 16]. Therefore, our aim is to develop a framework for generating synthetic medical images that not only exhibit visual realism but also demonstrate biological plausibility and *alignment* with clinical expertise.

One way to generate synthetic medical images is to finetune a pretrained "foundation" vision model, such as Stable Diffusion, that has been trained on billions of natural images (such as the LAION-5B dataset [17]), using real medical images. With a sufficiently large set of medical images, we can expect the finetuned model to capture the clinical knowledge encoded in medical images. However, the sample sizes of annotated medical images are typically limited to a few thousand. When a large vision model is finetuned on such a dataset using generic objective functions (such as the likelihood function), the model may capture only the generic features that make the medical images appear visually realistic, but it may miss nuanced features that make them biologically plausible and compliant with clinical domain knowledge (see examples in the next Section). Designing domain-specific objective functions for finetuning that ensure a generative model adheres to clinical knowledge is challenging. The difficulty arises from the numerous unpredictable ways in which these models can generate images that lack clinical plausibility. As a result, it is impractical to anticipate every possible failure scenario and manually construct a loss function that penalizes such instances.

**Summary of contributions.** In this paper, we develop a pathologist-in-the-loop framework for synthesizing medical images that align with clinical knowledge. Our framework is motivated by the success of reinforcement learning with human feedback (RLHF) in aligning the outputs of large language models (LLMs) with human preference [18, 19], and is directly inspired by [20], where human feedback was used to align the visual outputs of a generative model with input text prompts. To generate clinically-plausible medical images, our framework (outlined in Figure 1) comprises 3 steps:

***Step 1:*** We train a (conditional) diffusion model using real medical images. We then sample a synthetic dataset from the model to be evaluated by a pathologist. Each image is carefully examined, and the pathologist provides feedback on whether it meets the necessary criteria for clinical plausibility.

***Step 2:*** We collate a dataset of synthetic images paired with pathologist feedback and train a reward model to predict the pathologist feedback, i.e., clinical plausibility, on new images.

***Step 3:*** Finally, the reward model in Step 2 is utilized to incorporate expert knowledge into the generative model. This is achieved by finetuning the diffusion model using a reward-weighted loss function, which penalizes the generation of images that the pathologist considers clinically implausible.

Throughout this paper, we apply the steps above to the synthetic generation of bone marrow image patches, but the same conceptual framework can generalize to any medical imaging modality. We gathered pathologist feedback on thousands of synthetic images of various cell types generated by a conditional diffusion model. Then, we analyzed the impact of this feedback on the quality of the finetuned synthetic images. Our findings suggest that incorporating pathologist feedback significantly enhances the quality of synthetic images in terms of all existing quality metrics such as fidelity, accuracy of downstream predictive models, and clinical plausibility as evaluated by experts. Additionally, it also improves qualities that are not directly addressed in the pathologist evaluation, such as the diversity of synthetic samples. Furthermore, we show that human feedback can teach the generative model new clinical concepts, such as more refined identification of cell types, that are not annotated in the original training data. These results demonstrate the value of incorporating human feedback in clinical applications where generative models may not be readily suited to capture intricate and extensive clinical domain knowledge from raw data alone.

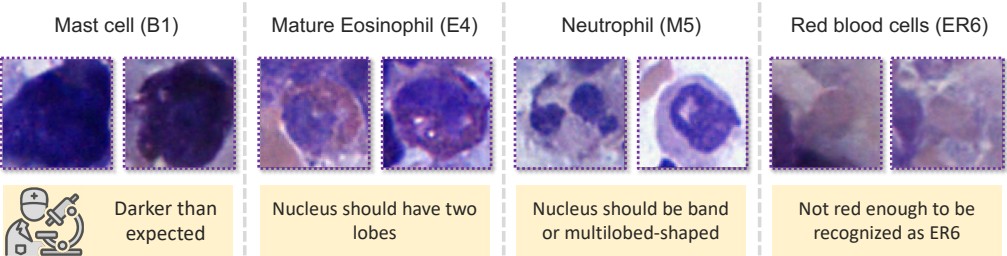

Figure 2: Samples for biologically-implausible synthetic bone marrow image patches across four different cell types. On the bottom panels, we show pathologist evaluations detailing the reasons for their implausibility.

## 2    Pathologist-in-the-Loop Generation of Synthetic Medical Images

In this Section, we provide a detailed description of our synthetic medical image generation framework. We will use a running example pertaining to single-cell images extracted from bone marrow aspirate whole slide images. Details of this setup and the dataset used in our study are provided in Section 4.

### 2.1    Step 1: Pathologist feedback collection

**Generative model pretraining.** The first step in our framework starts with training a generative model to synthesize medical images through the standard training procedure. As we discuss in more detail in Section 4, we utilized a dataset of 2,048 bone marrow image patches to train a conditional diffusion model [6]. The model was trained to generate class-conditional images, where an image class corresponds to a cell type. We conducted an exploratory analysis where we found that neither latent diffusion nor Stable Diffusion models yielded superior results compared to a customized diffusion model that we employed in this study (See Appendix A). We opted for using a class-conditional model rather than a text-conditional model as we found that existing pretrained vision-language models were not fit for capturing the scientific jargon related to bone marrow cell types.

Given a dataset of real images $\mathcal{D}_r = \{(x^i, c^i)\}_{i=1}^{n_r}$, where $x$ is a medical image and $c \in \mathcal{C}$ is an image class, we train a diffusion model to generate class-conditional images through the forward process:

$$x_{t+1} = x_t - \alpha_t \cdot \nabla_x \log(p_\theta(x_t|x, c)) + \epsilon_t, \tag{1}$$

where $\epsilon_t \sim \mathcal{N}(0, \rho^2)$ is the noise term at time-step $t$, $x_t$ is the data point at time-step $t$, $\alpha_t$ is the step size at $t$, $\theta$ is the model parameters and $\nabla_x \log(p(x_t|x, c))$ is the gradient of the log probability distribution with respect to $x$, conditioned on the original data $x$ and class $c$. Once the model is pretrained, we sample a synthetic dataset $\mathcal{D}_s = \{(\widetilde{x}^j, \widetilde{c}^j)\}_{j=1}^{n_s}$ by first sampling a class $\widetilde{c}$ from $\mathcal{C}$, and then sampling a medical image $\widetilde{x}$ conditioned on the class $\widetilde{c}$ through the reverse diffusion process.

**Pathologist evaluation.** Each image in the synthetic dataset $\mathcal{D}_s = \{(\widetilde{x}^j, \widetilde{c}^j)\}_{j=1}^{n_s}$ generated by the pre-trained model is inspected by an expert pathologist to assess its clinical plausibility. The objective of this evaluation is to identify the specific inaccuracies in the synthetic data that can only be identified by an expert, and provide feedback for the model to refine its synthesized images in the finetuning step.

When a model is trained with only a modest number of real image samples, it may generate bone marrow image patches that look visually appealing but are not biologically plausible. In Figure 2, we present 8 synthetic images sampled from the conditional diffusion model, which correspond to four different cell types. Each of these images achieves high precision and fidelity scores individually, but they also have biological implausibilities such as inaccurate cell coloring or nucleus shapes. Therefore, models that prioritize visual features without considering biological knowledge may miss important clinical features required for synthetic images to be useful for downstream analysis. Generic evaluation scores (e.g., [21, 22]) cannot diagnose these failures because they also lack biological domain knowledge. By incorporating feedback from pathologists, we can refine the generative model by identifying biological information that is missed by the pretrained model.

The expert pathologist examined each synthetic image and provided a feedback score on its biological plausibility. The evaluation typically involved inspecting the image and checking 7 aspects that contribute to its perceived plausibility (Table 1). These aspects pertain to the consistency of the shapes, sizes, patterns and colors of the contents of a synthetic bone marrow image $\widetilde{x}$ with the cell type $\widetilde{c}$.

| **Clinical plausibility criteria** | |
| --- | --- |
| (1) Cell size | (5) Chromatin pattern |
| (2) Nucleus shape and size | (6) Inclusions |
| (3) Nucleus-to-cytoplasm ratio | (7) Granules |
| (4) Cytoplasm color and consistency | |

Table 1: Pathologist evaluation criteria.

Among the determinants of plausibility is the cell size—different cells have different sizes, e.g., Lymphocytes are generally smaller than Monocytes or Neutrophils. Nucleus shape and size also depend on the cell type, e.g., Band Neutrophils have a horseshoe-shaped nucleus, whereas Segmented Neutrophils have a multi-lobed nucleus. Chromatin patterns within the nucleus are dense and clumped in Lymphocytes, while in Myeloid cells they are diffuse and fine. The number, size and color of granules also contribute to plausibility. Detailed explanation of all criteria is provided in the Appendix.

Note that we have full control over the number of synthetic images $n_s$, i.e., we can sample an arbitrary number of synthetic images from the conditional diffusion model. The key limiting factor on $n_s$ is the time-consuming nature of the feedback collection process. To enable scalable feedback collection, we limited our study to binary feedback, i.e., the pathologist flagged a synthetic image as "implausible" if they found a violation of any of the criteria in Table 1 upon visual inspection. We collected these binary signals and did not pursue a full checklist on all plausibility criteria for each synthetic image.

The output of Step 1 is an annotated dataset $\mathcal{D}_s = \{(\widetilde{x}^j, \widetilde{c}^j, \widetilde{y}^j)\}_{j=1}^{n_s}$, where $\widetilde{y}^j \in \{0, 1\}$ is the pathologist feedback on the $j$-th synthetic image, where $\widetilde{y}^j = 1$ means that the image is implausible.

## 2.2 Step 2: Clinical plausibility reward modeling

We conceptualize the pathologist as a "labelling function" $\Gamma : \mathcal{X} \times \mathcal{C} \to \{0, 1\}$ that maps the observed synthetic image $\widetilde{x}$ and declared cell type (class) $\widetilde{c}$ to a binary plausibility score. In Step 2, we model the "pathologist" by learning their labelling function $\Gamma$ on the basis of their feedback annotations.

To train a model $\Gamma$ that estimates the pathologist labelling function, we construct a training dataset that comprises a mixture of real and synthetic images as follows:

**Synthetic images**

We construct a dataset $\mathcal{D}_s^{\Gamma} = \{(\widetilde{x}^j, \widetilde{c}^j, \widetilde{y}^j)\}_{j=1}^{n_s}$ comprising the synthetic images and corresponding pathologist feedback collected in Step 1.

**Real images**

We build a dataset $\mathcal{D}_r^{\Gamma} = \{(x^i, \widetilde{c}^i, y^i)\}_{i=1}^{n_r}$ comprising the real images and pseudo-labels defined as $y^i = \mathbf{1}\{c^i \neq \widetilde{c}^i\}$, where $\widetilde{c}^i \sim \text{Uniform}(1, 2, \ldots, |\mathcal{C}|)$.

We combine both datasets $\mathcal{D}^{\Gamma} = \mathcal{D}_r^{\Gamma} \cup \mathcal{D}_s^{\Gamma}$ to construct a training dataset for the model $\Gamma$. The real dataset $\mathcal{D}_r^{\Gamma}$ is built by randomly permuting the image class and assigning an implausibility label of 1 if the permuted class does not coincide with the true class. We use the real dataset to augment the synthetic dataset with the annotated pathologist feedback. By augmenting the datasets $\mathcal{D}_r^{\Gamma}$ and $\mathcal{D}_s^{\Gamma}$, we teach the model $\Gamma$ to recognize two forms of implausibility in image generation: (i) instances where the synthetic image looks clinically plausible but belongs to a wrong cell type (i.e., training examples

in $\mathcal{D}_r^\Gamma$), and (ii) instances where the synthetic image is visually consistent with the correct cell type but fails to meet some of the plausibility criteria in Table 1 (i.e., subset of the training examples in $\mathcal{D}_s^\Gamma$). We call the resulting model $\Gamma$ a clinical plausibility *reward* model. Using the augmented feedback dataset $\mathcal{D}^\Gamma$, we train the reward model by minimizing the mean square error as follows:

$$L_\Gamma(\phi) = \sum_{j \in \mathcal{D}_s^\Gamma} (\widetilde{y}^j - \Gamma_\phi(\widetilde{x}^j, \widetilde{c}^j))^2 + \lambda_r \sum_{i \in \mathcal{D}_r^\Gamma} (y^i - \Gamma_\phi(x^i, \widetilde{c}^i))^2, \tag{2}$$

where $\lambda_r$ is a hyper-parameter that controls the contribution of real images in training the reward functions, and $\phi$ is the parameter of the reward model.

## 2.3 Step 3: Clinically-informed finetuning

In the final step, we refine the diffusion model by leveraging the pathologist feedback. Specifically, we incorporate domain knowledge into the model by utilizing the reward model $\Gamma$ in the finetuning objective. Following [20], we use a *reward-weighted* negative log-likelihood (NLL) objective, i.e.,

$$L(\theta, \widehat{\phi}) = \mathbb{E}_{(\widetilde{x}, \widetilde{c}) \sim \mathcal{D}_s} [-\Gamma_{\widehat{\phi}}(\widetilde{x}, \widetilde{c}) \cdot \log(p_\theta(\widetilde{x}|\widetilde{c}))] + \beta_r \cdot \mathbb{E}_{(x,c) \sim \mathcal{D}_r} [-\log(p_\theta(x|c))], \tag{3}$$

to finetune the conditional diffusion model, where $\beta_r$ is a hyper-parameter and $\widehat{\phi}$ is the reward model parameters obtained by minimizing (2) in Step 2. The finetuning objective in (3) incorporates the pathologist knowledge through the reward model, which predicts the pathologist evaluation of the synthetic images that the model generates as it updates its parameters $\theta$. The reward-weighted objective penalizes the generation of images that do not align with the pathologist preferences, hence we expect that the finetuned model will be less likely to generate clinically implausible synthetic images.

## 2.4 Bonus step: Feedback-driven generation of new clinical concepts

Besides refining generative models for clinical plausibility, pathologist feedback can be used to incorporate novel clinical concepts into the generative process that were not initially labeled in the real dataset. This could allow generative models to continuously adapt in changing clinical environments. For instance, in our bone marrow image generation setup, pathologist feedback can refine image generation by introducing new sub-types of the original cell types in $\mathcal{C}$, as illustrated in Figure 3.

Instead of collecting pathologist feedback that is limited to clinical plausibility, we also collect their annotation of new cell sub-types (e.g., segmented and band variants of Neutrophil cell types). Next, we train an auxiliary model $\Gamma_\rho(x)$ with parameter $\rho$ to classify the new sub-types based on the pathologist annotations. Finally, we finetune the conditional diffusion model through a combined loss function, i.e., $L(\theta, \widehat{\phi}) + L(\theta, \widehat{\rho})$, that incorporates the two forms of pathologist feedback, i.e., annotations of new cell types and clinical plausibility.

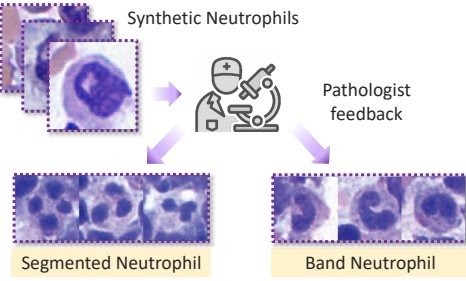

Figure 3: Generation of refined cell sub-types.

## 2.5 Pathologist feedback vs. Automated feedback

To asses the added value of human feedback, we consider a baseline where the generative model is supplemented with automatically generated feedback on clinical plausibility. To this end, we implement a baseline based on classifier-guided diffusion [23], where a classifier serves as an automatic feedback signal that deems a synthetic image implausible if it does not match the corresponding cell type. For this baseline, we train an auxiliary classifier to predict the cell type $c$ of an image $x$, and then we incorporate the gradient of the log-likelihood of this classifier in the training objective as described in [23] (See the Supplementary material for implementation details).

## 3 Related Work

Before delving into the experimental results, we discuss three strands of literature that are related to our synthetic data generation framework. These include: generative modeling for synthetic clinical data, evaluation of generative models and learning from human feedback.

**Generative modeling of synthetic medical images.** The dominant approach for synthesizing medical images is to train or finetune a generative model, such as a Variational Autoencoder (VAE) [24], a Generative Adversarial Network (GAN) [25], or a diffusion model [6, 23], using a sufficiently large sample of images from the desired modality. Owing to their recent success in achieving state-of-the-art results in high-fidelity image synthesis [23], diffusion probabilistic models have become the model of choice for medical image synthesis applications [12, 26–29]. In [12], the Stable Diffusion model—an open-source pretrained diffusion model—was used to generate synthetic X-ray images, and in [28] it was shown that diffusion models can synthesize high-quality Magnetic Resonance Images (MRI) and Computed Tomography (CT) images. [27] used latent diffusion models to generate synthetic images from high-resolution 3D brain images. All of these models are trained with the standard likelihood objective and the synthetic images are typically evaluated through downstream classification tasks or generic, domain-agnostic metrics for image fidelity. To the best of our knowledge, none of the previous studies have explored a human-AI collaboration approach to synthetic image generation or incorporated clinical knowledge into generative models of medical images.

**Evaluation of synthetic images in the medical domain and beyond.** Unlike discriminative modeling (i.e., predictive modeling) where model accuracy can be straightforwardly evaluated by comparing the model predictions with ground-truth labels in a testing set, evaluating the quality of generative models can be quite challenging since we do not have a "ground-truth" for defining what makes a synthetic sample is of high or low quality. Devising a generic score to evaluate a generative model can be tricky since there are many potential modes of failure [22]. Consequently, it is essential to design robust multidimensional scores that capture the most relevant failure modes for a given application.

Recently, there have been various attempts at defining domain-specific scores [30–32] as well as generic scores for evaluating the quality of synthetic images. Examples include the FID score which is based on a distributional distance between real and synthetic images [33]. Other examples for sample-level evaluation metrics include the precision and recall metrics [21] which check if synthetic data resides in the support of the real data distribution. However, these scores do not encode clinical domain knowledge, which is critical for identifying failures in generating clinically meaningful images. Traditional scores of medical image quality include signal- and contrast-to-noise ratio [34–36], mean structural similarity [37]. These scores are typically applied to real images and cannot be repurposed to judge the generative capacity of a synthetic data model in a meaningful way. The lack of an automated score for detecting clinically implausible synthetic medical images is a key motivation for our work. We believe that the most reliable way to assess the quality of a synthetic image is to have it evaluated by an expert pathologist. From this perspective, the reward model $\Gamma$ in Section 2.2 can be thought of as a data-driven score for image quality trained using pathologist evaluations.

**Learning from human feedback.** The success of many modern generative models can be attributed in part to finetuning using feedback solicited from human annotators. The utilization of human feedback in model finetuning is very common in natural language processing applications, particularly in finetuning of large language models (LLMs). Examples for applications were human feedback was applied include translation [38], web question-answering [39] and instruction tuning [40–42]. The key idea in these applications is that by asking a human annotator to rate different responses from the same model, one can use such annotations to finetune the model to align with human preference. Similar ideas have been applied to align computer vision models with human preferences [20, 43–46]. In the context of our application, the goal is to align the outputs of a generative model with the preferences of pathologists, which are naturally aligned with clinical domain knowledge. Our finetuning objective builds on the recent work in [20] and [45], which use human feedback to align text-prompts with generated images using a reward-weighted likelihood score.

## 4 Experiments

In this Section, we conduct a series of experiments to evaluate the utility of pathologist feedback in improving the quality of synthetic medical images. In the next Subsection, we start by providing a detailed description of the single-cell bone marrow image dataset used in our experiments.

### 4.1 Bone marrow cells dataset

In all experiments, we used a dataset of hematopathologist consensus-annotated single-cell images extracted from bone marrow aspirate (BMA) whole slide images. The images were obtained from the

| Morphological cell type | Code | Real data | | Synthetic data | Pathologist feedback | |
|---|---|---|---|---|---|---|
| | | Training | Testing | | Plausible | Implausible |
| Mast Cell | B1 | 128 | 32 | 213 | 72 | 141 |
| Basophil | B2 | 128 | 32 | 214 | 29 | 185 |
| Immature Eosinophil | E1 | 128 | 32 | 213 | 49 | 164 |
| Mature Eosinophil | E4 | 128 | 32 | 224 | 53 | 171 |
| Pronormoblast | ER1 | 128 | 32 | 256 | 97 | 159 |
| Basophilic Normoblast | ER2 | 128 | 32 | 256 | 49 | 207 |
| Polychromatophilic Normoblast | ER3 | 128 | 32 | 256 | 129 | 127 |
| Orthochromic Normoblast | ER4 | 128 | 32 | 256 | 118 | 138 |
| Polychromatophilic Erythrocyte | ER5 | 128 | 32 | 256 | 141 | 115 |
| Mature Erythrocyte | ER6 | 128 | 32 | 256 | 120 | 136 |
| Myeloid Blast | M1 | 128 | 32 | 256 | 138 | 118 |
| Promyelocyte | M2 | 128 | 32 | 256 | 107 | 149 |
| Myelocyte | M3 | 128 | 32 | 256 | 131 | 125 |
| Metamyelocyte | M4 | 128 | 32 | 256 | 88 | 168 |
| Mature Neutrophil | M5 | 128 | 32 | 256 | 80 | 176 |
| Monocyte | MO2 | 128 | 32 | 256 | 83 | 173 |

Table 2: Breakdown of bone marrow image patches by morphological cell type and pathologist feedback.

clinical archives of an academic medical center. The dataset comprised 2,048 images, with the images evenly distributed across 16 morphological classes (cell types), with 160 images per class (Table 2). These classes encompass varied cell types found in a standard bone marrow differential. The dataset covers the complete maturation spectrum of Erythroid and Neutrophil cells, from Proerythroblast to mature Erythrocyte and from Myeloid blast to mature Neutrophils, respectively. The dataset also differentiates mature Eosinophils with segmented nuclei from immature Eosinophils and features Monocytes, Basophils, and Mast cells. Bone marrow cell counting and differentiating between various cell types is a complex task that poses challenges even for experienced hematologists. Hence, we expect pathologist feedback to significantly improve the quality of bone marrow image synthesis.

## 4.2    Synthetic data generation and pathologist feedback collection

We used a conditional diffusion model trained on real images ($64 \times 64$ pixels in size) to generate synthetic image patches. Training was conducted using 128 images per cell type, with 32 images per cell type held out for testing and evaluating all performance metrics. For the reward model $\Gamma(x, c)$, we used a ResNeXt-50 model [47] pre-trained on a cell type classification task to obtain embeddings for individual images, and then concatenated the embeddings with one-hot encoded identifiers of image class (cell type) as inputs to a feed-forward neural network that predicts clinical plausibility. Further details on model architectures and selected hyper-parameters are provided in the Appendix.

We collected **feedback** from an expert pathologist on **3,936 synthetic images** generated from the diffusion model. The pathologist identified most of these images as implausible—the rate of implausible images was as high as 85% for some cell types (e.g., Basophil cells, see Table 2). After training the reward model using pathologist feedback, we finetune the diffusion model as described in Section 2.

## 4.3    Results

**Expert evaluation of synthetic data quality.** To evaluate the impact of pathologist feedback on the generated synthetic data, we created two synthetic datasets: a sample from the diffusion model (before finetuning with pathologist feedback) and a sample from the finetuned version of the model after incorporating the pathologist feedback. Each dataset comprised 400 images (25 images per cell type). An expert pathologist was asked to evaluate the two samples and label each image as plausible or implausible (in a manner similar to the feedback collection process).

Table 3 lists the fraction of clinically plausible images per cell type for the two synthetic datasets (before and after finetuning using the pathologist feedback) as evaluated by an expert hematopathologist. As we can see, the pathologist feedback leads to a significant boost in the quality of synthetic images across all cell types, increasing the average rate of clinical plausibility from 0.21 to 0.75. Note that in this experiment, the human evaluator emulates the reward function $\Gamma(x, c)$. Hence, the improved performance of the finetuned model indicates success in learning the pathologist preferences.

**Evaluating synthetic data using fidelity & diversity scores.** In addition to expert evaluation, we also evaluated the two synthetic datasets generated in the previous experiment using standard metrics for evaluating generative models. We considered the Precision, Recall and Coverage metrics [21, 22].

| | % Clinically plausible | |
|---|---|---|
| | *No feedback* | *Path. feedback* |
| B1 | 0.40 | 0.92 |
| B2 | 0.16 | 1.00 |
| E1 | 0.12 | 0.80 |
| E4 | 0.24 | 0.52 |
| ER1 | 0.44 | 0.96 |
| ER2 | 0.28 | 0.64 |
| ER3 | 0.24 | 0.68 |
| ER4 | 0.20 | 0.84 |
| ER5 | 0.20 | 0.76 |
| ER6 | 0.20 | 0.96 |
| M1 | 0.20 | 0.84 |
| M2 | 0.20 | 0.64 |
| M3 | 0.08 | 0.56 |
| M4 | 0.20 | 0.84 |
| M5 | 0.08 | 0.68 |
| MO2 | 0.12 | 0.40 |

Table 3: Expert evaluation of synthetic data.

| Training data | Precision | Recall | Coverage |
|---|---|---|---|
| Synthetic (no feedback) | 68.06 | 52.00 | 56.98 |
| Synthetic (auto. feedback) | 74.80 | 43.90 | 61.33 |
| Synthetic (with feedback) | **81.01** | **56.74** | **84.57** |

Table 4: Evaluation of synthetic data using fidelity and diversity metrics.

| Training data | F1 | Accuracy | Precision | Recall |
|---|---|---|---|---|
| Synthetic (no feedback) | 60.33 | 95.17 | 61.33 | 65.56 |
| Synthetic (auto. feedback) | 63.47 | 95.58 | 64.64 | 65.68 |
| Synthetic (with feedback, $\mathcal{D}_s^\Gamma$) | 71.80 | 96.41 | 71.29 | 74.51 |
| Synthetic (with feedback, $\mathcal{D}^\Gamma$) | 75.80 | 96.95 | 75.59 | 76.51 |
| **Real** | **79.03** | **97.39** | **79.10** | **79.47** |

Table 5: Accuracy of classifiers trained on real and synthetic data.

Precision measures the fraction of synthetic samples that resides in the support of the real data distribution and is used to measure fidelity. Recall and Coverage measure the "diversity" of synthetic samples, i.e., the fraction of real images that are represented in the output of a generative model. In addition to the two synthetic datasets (with and without feedback) generated in the previous experiment, we also evaluated a third synthetic dataset finetuned using the automatic feedback (classifier-guidance) approach described in Section 2.5. The results in Table 4 show that pathologist feedback improves the quality of synthetic data compared to the two baselines across all metrics under consideration. Interestingly, we see that feedback not improves fidelity of synthetic images, but it also improves the diversity of samples, which was not one the criteria considered in the pathologist feedback.

**Downstream modeling with synthetic data.** Morphology-based classification of cells is a key step in the diagnosis of hematologic malignancies. We evaluated the utility of synthetic medical images in training a cell-type classification model. In this experiment, we train a ResNext-50 model to classify the 16 cell types using real data, synthetic data from the pretrained model with no feedback, and synthetic data from the model finetuned with pathologist feedback. To ensure a fair comparison, we created synthetic datasets consisting of 128 images per cell type, matching the size of the real data. The classification accuracy of all models was then tested on the held-out real dataset, containing 32 images per cell type. Results are shown in Table 5. Unsurprisingly, the model trained on real data demonstrated superior performance across all accuracy metrics, exhibiting a significant gap compared to the model trained on synthetic data without human feedback. However, incorporating pathologist feedback helped narrow this gap and improved the quality of synthetic data to the point where the resulting classifiers only slightly underperformed compared to the one trained on real data.

To evaluate the marginal value of human feedback, we also considered two ablated versions of our synthetic data generation process. These included a synthetic dataset generated using automatic feedback (Section 2.5) as well as a reward model trained using the pathologist feedback on synthetic data only (i.e., dataset $\mathcal{D}_s^\Gamma$) without real data augmentation (Section 2.2). The results in Table 5 show that automatic feedback only marginally improves performance, which aligns with the results in Table 4. Real data augmentation (i.e., finetuning on $\mathcal{D}_s^\Gamma + \mathcal{D}_r^\Gamma$) slightly improves classification accuracy, but most of the performance gains are achieved by finetuning on the pathologist-labeled dataset $\mathcal{D}_s^\Gamma$.

**Impact of the number of feedback points.** How much human feedback is necessary to align the generative model with the preferences of pathologists? In Figure 4, we analyze the effect of different amounts of pathologist-labeled synthetic images on training the reward model. We explore four scenarios: 0%, 10%, 50%, and 100% of the 3,936 synthetic images labeled by the pathologist. For each scenario, we fine-tune the pretrained model using a reward model trained with the corresponding fraction of synthetic images. We then repeat the cell classification experiment to evaluate the accuracy of the classifiers trained using synthetic data generated in these four scenarios. Figure 4(a) shows that as the number of pathologist-labeled synthetic images increases, all accuracy metrics increase to improve over the pretrained model performance and become closer to the accuracy of training on real data. Additionally, we see that even a modest amount of feedback (e.g., 10% of pathologist-labeled synthetic images) can have a significant impact on performance. Figure 4(b) also show the qualitative improvement in the quality of synthetic images as the amount of human feedback increases.

**Incorporating new clinical concepts using pathologist feedback.** Finally, we evaluate the feedback-driven generation approach outlined in Section 2.4. Here, our goal is not only to leverage pathologist

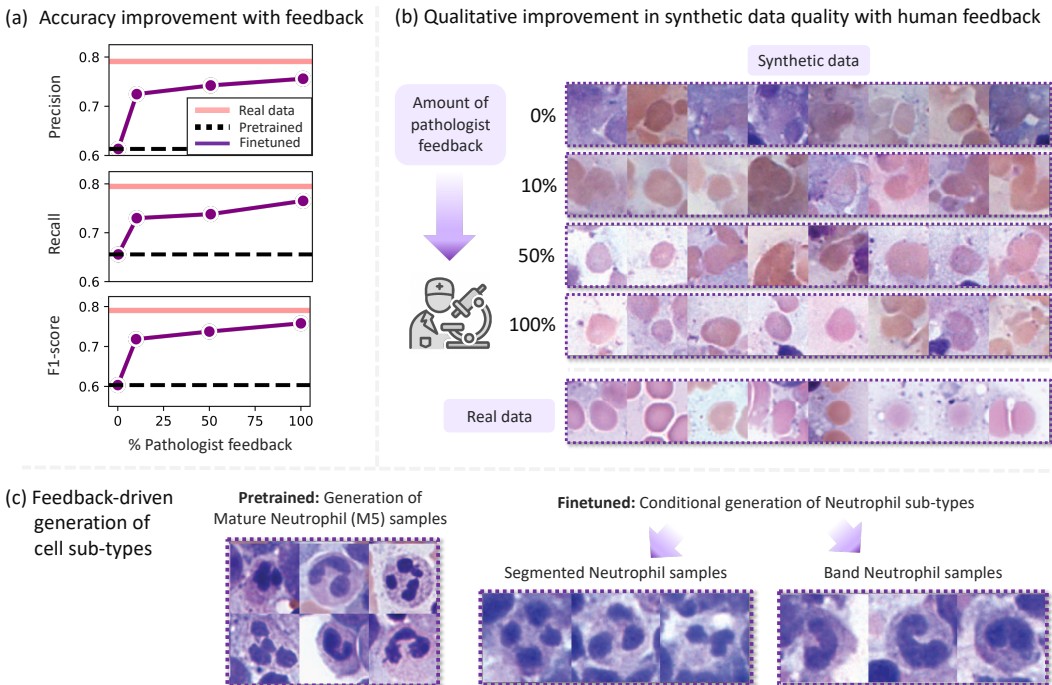

Figure 4: **Quantitative and qualitative impact of pathologist feedback on synthetic images.** (a) Accuracy of cell-types classifiers trained on synthetic data with varying amounts of pathologist feedback. (b) Visual inspection of random synthetic samples from diffusion models finetuned with with varying amounts of pathologist feedback. (c) Feedback-driven conditional generation of new (segmented and band) subtypes of Neutrophil cells.

feedback for rating the plausibility of synthetic images, but also to harness their expertise in providing additional annotations that can enable the conditional diffusion model to generate more refined categories of bone marrow image patches, e.g., abnormal cell types that develop from preexisting normal cell types. Hence, refining the generative model to synthesize new cell subtypes can help continuously update the model to capture new pathological cells and build downstream diagnostic models.

In this experiment, we focus on finetuning the model to distinguish between band Neutrophils and segmented Neutrophils (Figure 5(c)). These two sub-categories are often lumped together in bone marrow cell typing, as was the case in our dataset (see Table 2). However, in many clinical settings, differentiating between the two subtypes is essential. For instance, a high percentage of band Neutrophils can indicate an acute infection, inflammation, or other pathological conditions. We collected pathologist annotations of band and segmented Neutrophils, and trained a subtype classifier to augment the plausibility reward as described in Section 2.4. The finetuned model was able to condition on the new classes and generate plausible samples of the two Neutrophil subtypes (Figure 5(c)), while retaining the classification accuracy with respect to the new classes (See Appendix for detailed results).

## Conclusions

Synthetic data generation holds great potential for facilitating the sharing of clinical data and enriching rare diseases datasets. However, existing generative models and evaluation metrics lack the ability to incorporate clinical knowledge. Consequently, they often fall short in producing clinically plausible and valuable images. This paper introduces a pathologist-in-the-loop framework for generating clinically plausible synthetic medical images. Our framework involves finetuning a pretrained generative model through feedback provided by pathologists, thereby aligning the synthetic data generation process with clinical expertise. Through the evaluation of synthetic bone marrow patches by expert hematopathologists, leveraging thousands of feedback points, we demonstrate that human input significantly enhances the quality of synthetic images. These results underscore the importance of incorporating human feedback in clinical applications, particularly when generative models encounter challenges in capturing nuanced domain knowledge solely from raw data.

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

# A  Experimental Setup

## A.1  Exploratory analysis of generative models

To select the generative model underlying our pathologist-in-the-loop framework, we conducted an exploratory analysis for the performance of various models, including conditional GANs, a conditional diffusion model, and a latent diffusion model [1]. Following an evaluation of these models, we selected the conditional diffusion model as our baseline for incorporating human feedback. The decision to choose the conditional diffusion model as our baseline was based on the fact that it outperformed the other baselines when pretrained on the bone marrow image patches. The rationale for selecting the best performing model in the pre-training phase is that the marginal value of pathologist feedback is best assesed when incorporated into a model that already performs well without feedback.

We found that the conditional GAN model did not converge to a reliable set of parameters and was therefore deemed unfit as a base model in our framework. The latent diffusion model demonstrated the ability to generate visually appealing cell images, but a majority of these images did not correspond to the intended class labels. (See Table A.1 and Figure A.1 for a quantitative and qualitative comparison of all baseline generative models.) In addition to these generative baselines, we also finetuned the Stable Diffusion model on our dataset, but the resulting images displayed an artistic style that could not be identified by expert as realistic bone marrow slides.

Table A.1: Performance comparison for baseline generative models in the cell classification task.

| Training Data | Test Data | AUC | F1 | Precision | Recall |
|---|---|---|---|---|---|
| Conditional diffusion model | Real | 0.929482 | 0.567672 | 0.604237 | 0.604146 |
| Conditional GAN model | Real | 0.413551 | 0.003225 | 0.002186 | 0.015804 |
| Conditional Latent diffusion model | Real | 0.566388 | 0.008565 | 0.048372 | 0.051456 |

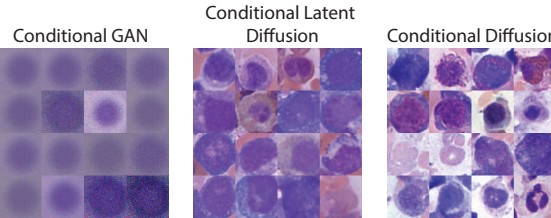

Figure A.1: Representative samples from different baseline models

## A.2  Dataset details, image acquisition and annotation

Our dataset was created from pathology slides for bone marrow aspirate. All slides were randomly selected from the clinical service based on normal morphology and adequate specimen. The slides were stained using a version of the Fisherbrand modified Wright-Giemsa Stain Pack. The quality of images varied across slides, reflecting variations in stain intensity, slide preparation, and slide age.

**Cell types included in our analysis.** The 16 image classes listed in Table 2 cover most cell types included in a standard bone marrow differential. A very broad spectrum of Erythroid and Neutrophil maturation was included, from Proerythroblast to mature Erythrocyte, and from Myeloid blast to Neutrophil. Eosinophils were separated into mature Eosinophils with segmented nuclei and immature Eosinophils. In addition, the set included Monocytes, Basophils and Mast cells. We extracted 128 images per class for training and 32 image per class for evaluating performance metrics.

**Pathologist annotation and criteria for clinical plausibility.** Clinicians consider various aspects when annotating the (real) bone marrow cell images, and these aspects also underlie the criteria for evaluating the plausibility of synthetic images (Table 1). These criteria include **(1) Cell size:** Cells can be classified based on their size, ranging from small to large. **(2) Nucleus shape and size:** The shape and size of a cell nucleus can provide valuable information about the cell type. For instance, band

Neutrophils have a horseshoe-shaped nucleus, while segmented Neutrophils have a multi-lobed nucleus. **(3) Nucleus-to-cytoplasm ratio:** The proportion of the cell occupied by the nucleus compared to the cytoplasm vary across cell types. Immature cells usually have a larger nucleus-to-cytoplasm ratio, while mature cells have a smaller ratio. **(4) Chromatin pattern:** The appearance of chromatin (DNA and proteins) within the nucleus can offer clues to the cell type. For example, Lymphocytes typically have dense, clumped chromatin, while Myeloid cells have a more diffuse, fine chromatin pattern. **(5) Cytoplasm color and consistency:** The color and texture of a cell cytoplasm can be indicative of its type. Erythroid cells have a deeply Basophilic (blue) cytoplasm, while Myeloid cells have a lighter, Eosinophilic (pink) cytoplasm. **(6) Granules:** The presence, size, number, and color of granules within the cytoplasm can help identify cell types. Neutrophils have small, pale granules, Eosinophils have large, orange-pink granules, and Basophils have large, dark purple granules.

### A.3 Model architecture

**Diffusion model.** We used conditional diffusion [48] as our baseline generative model. Our fine-tuning pipeline is based on public repository (https://github.com/openai/improved-diffusion.git). The model was fine-tuned using the Adam optimizer [49]. The model is trained in half-precision on $2 \times$ 24 GB NVIDIA GPUs, with a per-GPU batch size of 16, resulting in a toal batch size of 32. We used a learning rate of 104, and an exponential moving average over parameters with a rate of 0.9999.

**Classification model.** To evaluate the utility of synthetic data in downstream tasks, we trained a classification model to classify cell types in synthetic data and tested its performance in real data. We implemented this classifier using the ResNeXt-50 architecture, which have been shown to display good performance in classifying bone marrow cells in [50]. (ResNeXt is a convolutional neural network architecture, which combines elements of of VGG, ResNet and Inception models.) The network was initialized using weights from ImageNet dataset and then trained using our dataset.

**Reward model.** The reward model $\Gamma(x, c)$ takes the image and its cell type as input, and issues a plausibility scores that predicts whether a pathologist would label the image as clinically plausible. The reward model was constructed as a feed-forward neural network that operates on the cell type as well as the embedding feature of images extracted from the ResNext network structure.

## B Performance Metrics for Generative Models

We implemented the evaluation scores for generative models in Section 4.3 based on publicly released repository (https://github.com/clovaai/generative-evaluation-prdc). The precision and recall scores, originally introduced in [21], are formally defined as:

$$\text{Precision} = \frac{1}{M} \sum_{j=1}^{M} 1_{\{Y_j \in \text{Manifold}(X_1, \ldots, X_N)\}}, \ \text{Recall} = \frac{1}{N} \sum_{j=1}^{N} 1_{\{X_j \in \text{Manifold}(Y_1, \ldots, Y_M)\}},$$

where the manifold is the defined as $\text{Manifold}(X_1, \cdots, X_N) := \bigcup_{i=1}^{N} B(X_i, \text{NND}_k(X_i))$, with $B(x, r)$ being a Euclidean ball around the point $x$ with radius $r$ and $\text{NND}_k(X_i)$ is the distance to the $k - th$ nearest neighbour. The coverage metric is defined as follows:

$$\text{Coverage} = \frac{1}{N} \sum_{i=1}^{N} 1_{\{\exists \, j \text{ s.t. } Y_j \in B(X_i, \text{NND}_k(X_i))\}}.$$

In the definitions above, $\{X_i\}_i$ denotes the real sample and $\{Y_j\}_j$ denotes the synthetic sample. The precision metric evaluates the fraction of synthetic samples that reside in the real data manifold, whereas the recall and coverage metrics evaluate the fraction of real samples that are represented in the synthetic sample. Precision can be though of as an automated score for clinical plausibility.

## C Incorporating New Clinical Concepts using Pathologist Feedback

In Section 4.3, we discussed the use of human feedback to provide additional annotations that can enable the conditional diffusion model to generate more refined categories of bone marrow image patches. In this experiment, we finetuned the model to distinguish between band Neutrophils and

segmented Neutrophils (Figure 5(c)). These two sub-categories are often lumped together in bone marrow cell typing, as was the case in our dataset (see Table 2). However, in many clinical settings, differentiating between the two subtypes is essential. For instance, a high percentage of band Neutrophils can indicate an acute infection, inflammation, or other pathological conditions. We collected pathologist annotations of band and segmented Neutrophils, and trained a subtype classifier to augment the plausibility reward as described in Section 2.4. The finetuned model was able to condition on the new classes and generate plausible samples of the two Neutrophil subtypes (Figure 5(c)), while retaining the classification accuracy with respect to the new classes as shown in Table C.2.

Table C.2: Model finetuned with the new subtypes of Neutrophil cells.

| AUC score | Accuracy | Precision | Recall |
|-----------|----------|-----------|--------|
| 81.90 | 71.88 | 67.95 | 82.81 |

## D   Pseudocodes

---
**Algorithm 1** Training the reward model
---
1: **function** REWARD_FUNCTION (Training from pathologist feedback)
2:     $img_{embedding} = NN.encode_image(preprocess(Image))$
3:     $class_{embedding} = NN.encode_label(label)$
4:     **return** $pred_feedback(img_{embedding}, class_{embedding})$
5: **end function**
6: **for** (Image, label, feedback) in Dataloader **do**
7:     scores = Reward_function(Image, label)
8:     loss = MSELoss(scores, feedback) optimizer.zero_grad()
       loss.backward() optimizer.step()
9: **end for**
---

---
**Algorithm 2** Pretraining the conditional diffusion model for generating synthetic images
---
       $D$, a dataset of real images $N$, the number of training iterations $B$, the batch size $\alpha$, learning rate A trained conditional diffusion model Initialize the conditional diffusion model $M$ $i = 1$ to $N$ Sample a mini-batch of images $X \subset D$ with size $B$ Compute the noise schedule $\sigma(t)$ for the diffusion process Initialize the set of generated images $Y$ with denoised images from $X$ $t = T$ downto 1 Compute the conditional diffusion probabilities $p(y_t|y_{t+1})$ for $Y$ using model $M$ Update $Y$ by sampling $y_t$ from $p(y_t|y_{t+1})$ Compute the reconstruction loss $L$ between the generated images $Y$ and the real images $X$ Update the model parameters $\theta$ using gradient descent: $\theta \leftarrow \theta - \alpha\nabla_\theta L$
---

**Algorithm 3** Finetuning the conditional diffusion model using pathologist feedback

$D_r$, a dataset of real images $D_a$, a dataset of artificial images generated from diffusion models trained on the real images $N$, the number of training iterations $B$, the batch size $\alpha$, learning rate $\gamma$, penalty parameter $Model_{Reward}$, Reward Model trained with clinician feedback Fine tuning Conditional Diffusion Model Load the trained conditional diffusion model $M$ $i = 1$ to $N$ Sample a mini-batch of images $X \subset D$ with size $B$ Compute the noise schedule $\sigma(t)$ for the diffusion process Initialize the set of generated images $Y$ with denoised images from $X$ $t = T$ downto 1 Compute the conditional diffusion probabilities $p(y_t|y_{t+1}, c)$ for $Y$ and $classC$ using model $M$ Update $Y$ by sampling $y_t$ from $p(y_t|y_{t+1})$ Compute the reconstruction loss $L$ between the generated images $Y$ and the original images $X$ $X \in D_r loss = loss * \gamma$ $X \in D_a$

$loss = loss * Model_{Reward}(X)$       RASIEERROR Update the model parameters $\theta$ using gradient descent: $\theta \leftarrow \theta - \alpha \nabla_\theta L$

---

**Algorithm 4** Incorporating new clinical concepts into the model

$D_{pre}$, a collection of artificial images, a mixture of sub types

1: **function** NEW_CONCEPT_FUNCTION (Training from clinician feedback)
2:     $img_{embedding} = NN.encode_i mage(preprocess(Image))$
3:     $class_{embedding} = NN.encode_l abel(label)$
4:     **return** $pred_f eedback(img_{embedding}, class_{embedding})$
5: **end function**
6: **for** (Image, label, feedback) in Dataloader **do**
7:     scores = Reward_function(Image, label)
8:     loss = MSELoss(scores, feedback) optimizer.zero_grad()
       loss.backward() optimizer.step()
9: **end for**

---

## E   Illustration of Synthetic Samples

In this Section, we present illustrative samples of synthetic bone marrow image patches generated by our model.

**Figure E.1.**  Representative samples from the conditional diffusion model before (left) and after (right) incorporating pathologist feedback.

**Figure E.2.** Representative samples from the conditional diffusion model.

**Figure E.3.** Representative samples from the finetuned model with 10% of the pathologist feedback points.

**Figure E.4.** Representative samples from the finetuned model with 50% of the pathologist feedback points.

**Figure E.5.** Representative samples from the finetuned model with 100% of the pathologist feedback points.

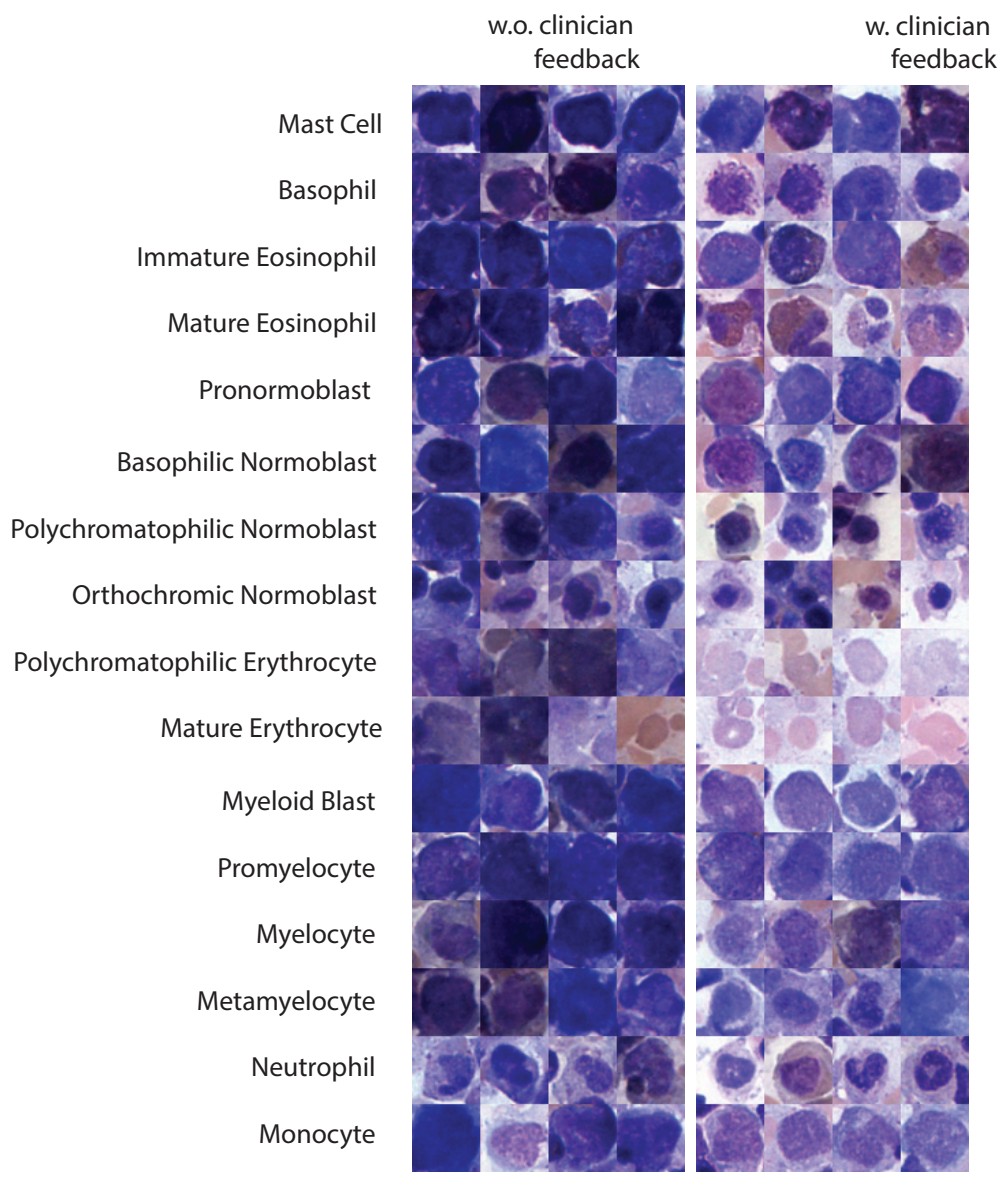

Figure E.1: Representative samples from the conditional diffusion model before (left) and after (right) incorporating pathologist feedback.

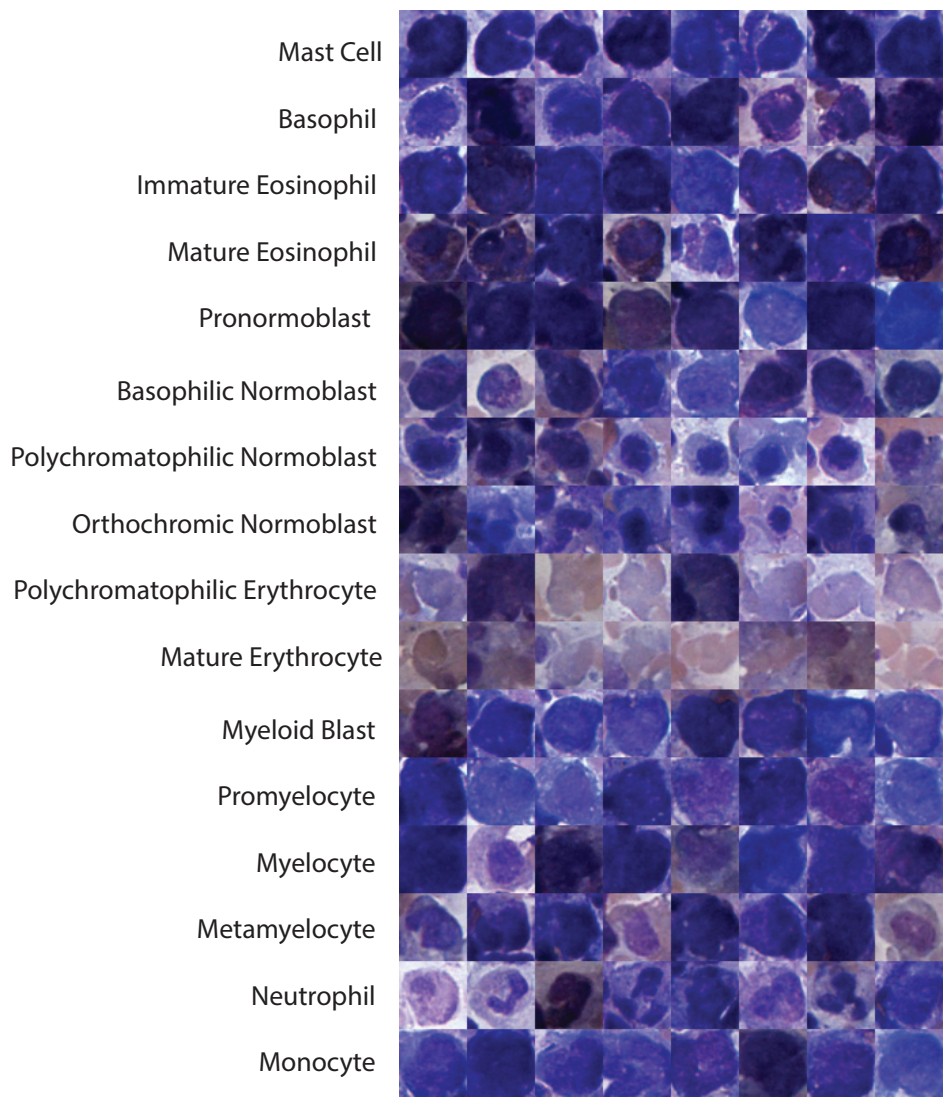

Mast Cell
Basophil
Immature Eosinophil
Mature Eosinophil
Pronormoblast
Basophilic Normoblast
Polychromatophilic Normoblast
Orthochromic Normoblast
Polychromatophilic Erythrocyte
Mature Erythrocyte
Myeloid Blast
Promyelocyte
Myelocyte
Metamyelocyte
Neutrophil
Monocyte

Figure E.2: Representative samples from the conditional diffusion model.

Mast Cell

Basophil

Immature Eosinophil

Mature Eosinophil

Pronormoblast

Basophilic Normoblast

Polychromatophilic Normoblast

Orthochromic Normoblast

Polychromatophilic Erythrocyte

Mature Erythrocyte

Myeloid Blast

Promyelocyte

Myelocyte

Metamyelocyte

Neutrophil

Monocyte

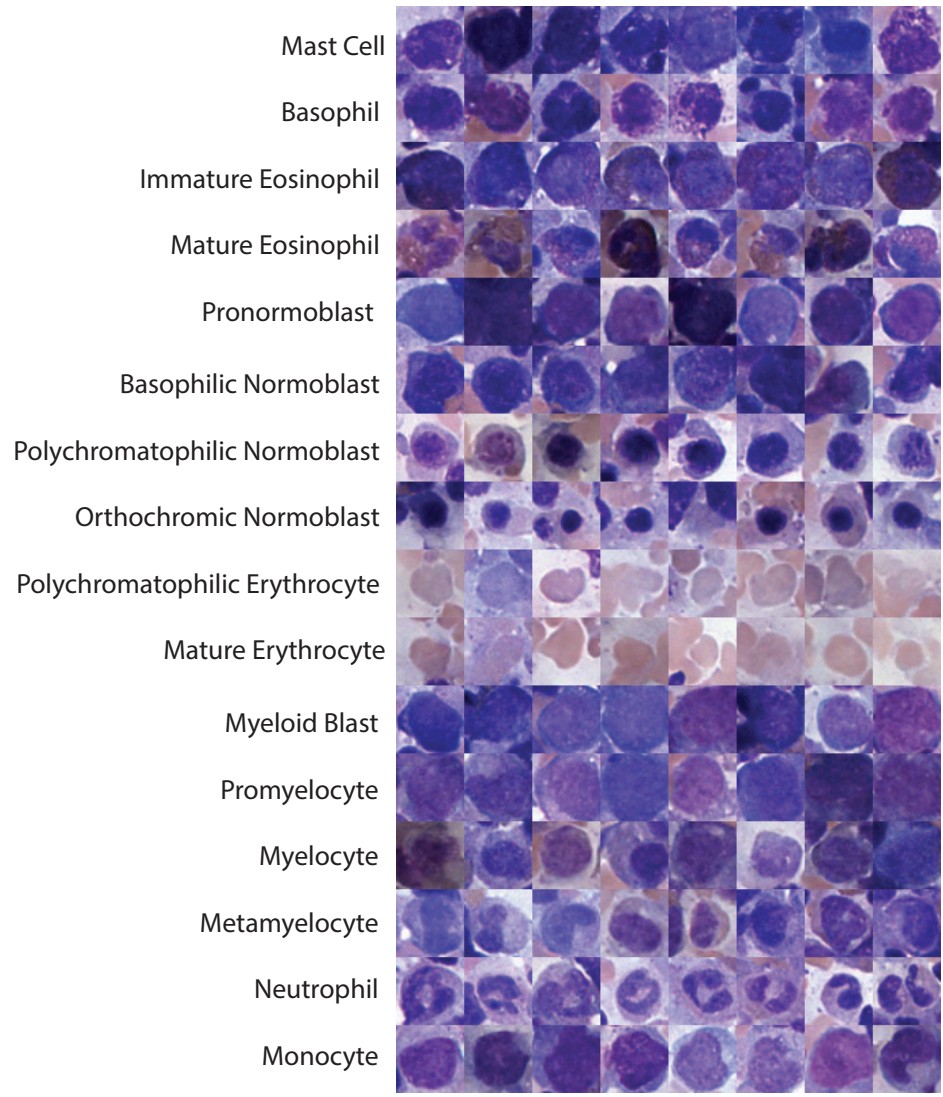

Figure E.3: Representative samples from the finetuned model with 10% of the pathologist feedback points.

Mast Cell

Basophil

Immature Eosinophil

Mature Eosinophil

Pronormoblast

Basophilic Normoblast

Polychromatophilic Normoblast

Orthochromic Normoblast

Polychromatophilic Erythrocyte

Mature Erythrocyte

Myeloid Blast

Promyelocyte

Myelocyte

Metamyelocyte

Neutrophil

Monocyte

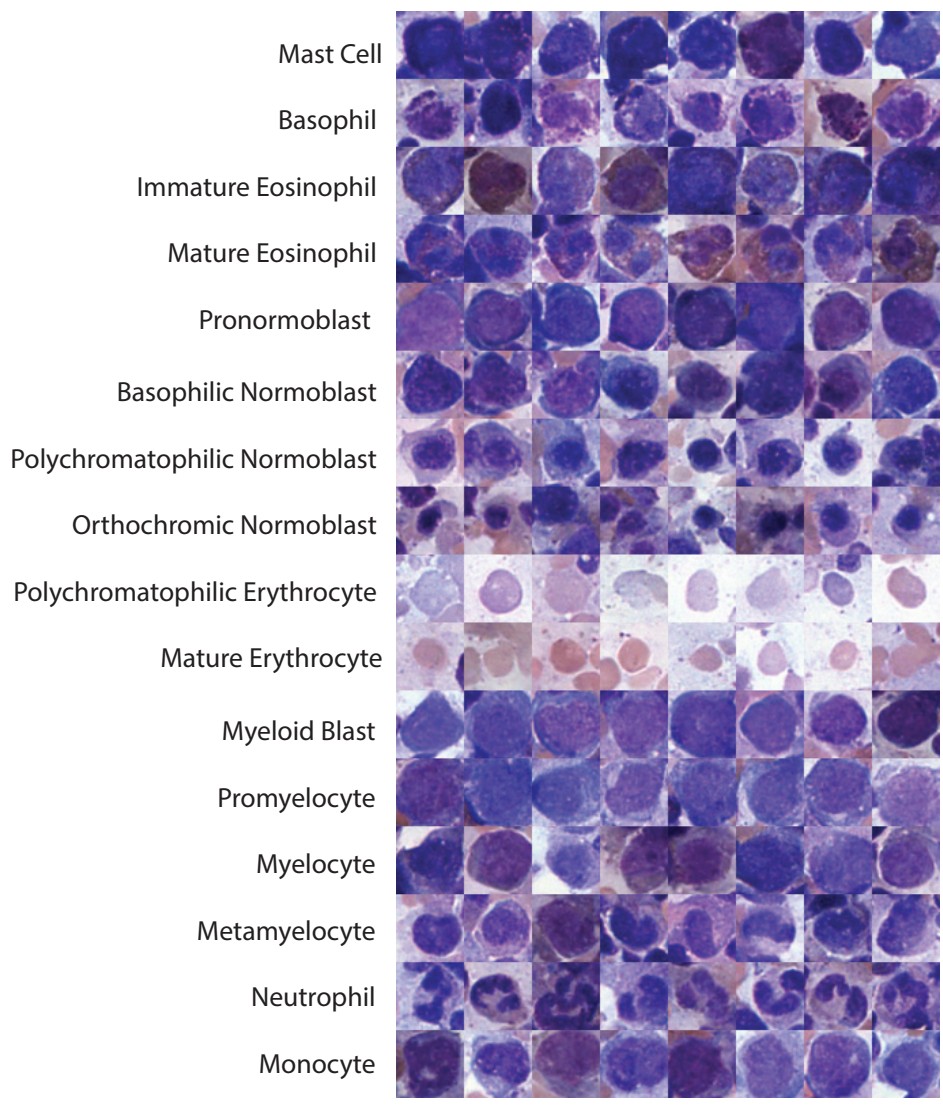

Figure E.4: Representative samples from the finetuned model with 50% of the pathologist feedback points.

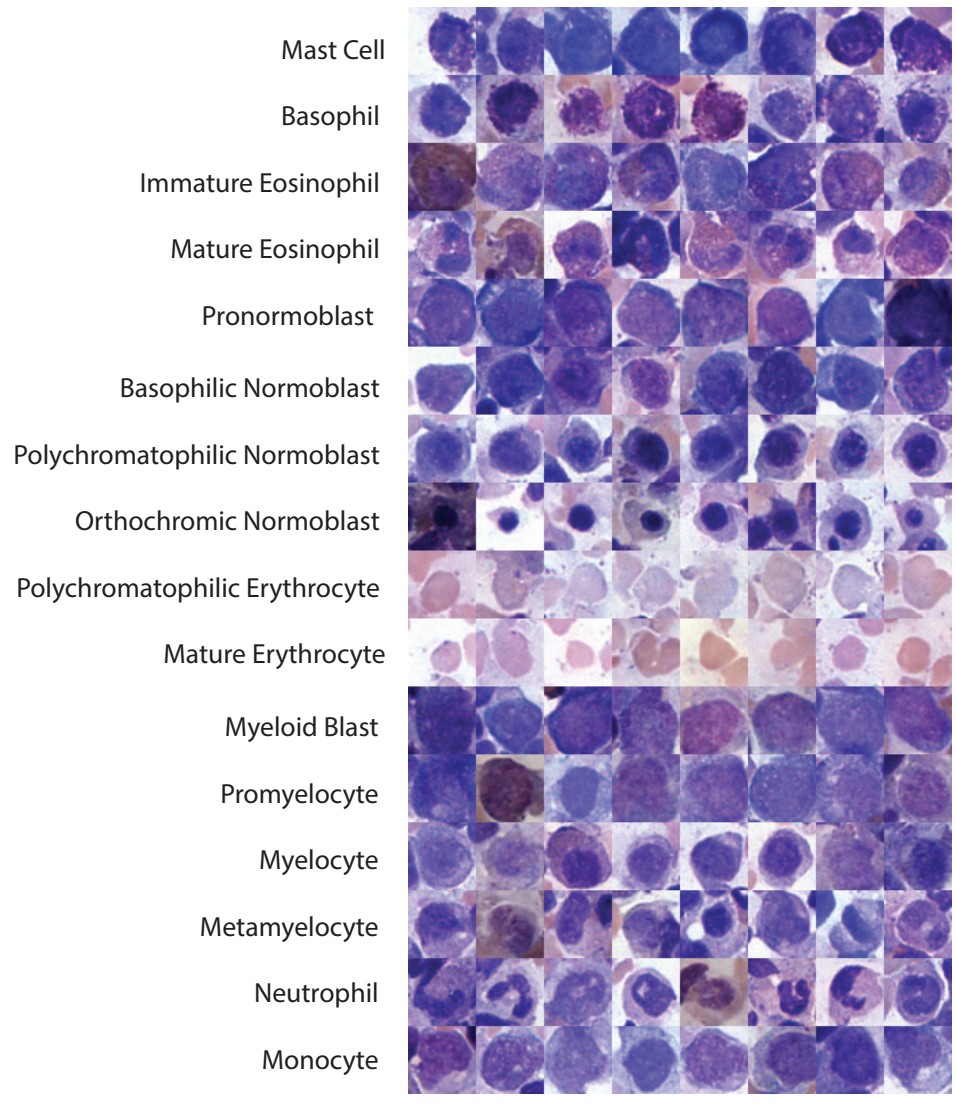

Figure E.5: Representative samples from the finetuned model with 100% of the pathologist feedback points.

Mast Cell
Basophil
Immature Eosinophil
Mature Eosinophil
Pronormoblast
Basophilic Normoblast
Polychromatophilic Normoblast
Orthochromic Normoblast
Polychromatophilic Erythrocyte
Mature Erythrocyte
Myeloid Blast
Promyelocyte
Myelocyte
Metamyelocyte
Neutrophil
Monocyte

