# OpenReview forum: "Aligning Synthetic Medical Images with Clinical Knowledge using Human Feedback"
_NeurIPS.cc/2023/Conference — NeurIPS 2023 spotlight_

### Official Review · Reviewer_Mvby · 2023-07-03

**Soundness:** 4 excellent
**Presentation:** 3 good
**Contribution:** 4 excellent
**Rating:** 6
**Confidence:** 5

**Summary:**

This paper proposed to use the diffusion model with expert feedback to improve the quality of synthetic medical images. The key motivation is the embedding of RLHF for medical image generation. To achieve this, there are three steps: (1) pre-trained a generation model and collected expert feedback based on several manual rules; (2) trained a model to predict the feedback scores; (3) incorporated the scores to fine-tune generative models. Extensive experiments demonstrate the effectiveness of the proposed model. Overall, this is an interesting and sound paper.

**Strengths:**

1. RLHF-based generation model for medical knowledge incorporation. Interesting and useful.
2. Reasonable framework. The proposed model is based on the conditional diffusion models and the whole process is sound.
3. Extensive evaluation. Besides the commonly-used quality-related metrics and downstream performance, the new knowledge discovery and visual results are also impressive.

**Weaknesses:**

1. Reproductivity. It's better to make the codes public and improve the reproductivity.
2. Manual rules. The expert feedback is highly related to the manual design rules. It's better to see the performance with different amounts of rules or with more rules. Also, please elaborate more on how to define these rules. In other words, it is not clear why the designed rules can indicate the complex image quality and small image differences.
3. Downstream performance. There is only one downstream task to denote effectiveness. More tasks should be considered. Meanwhile, why not show the performance of downstream models trained by both synthetic and real images?

**Questions:**

See the above weaknesses.

**Limitations:**

Besides the comments on the paper's weaknesses, please also consider:

1. Even though it is a kind of application paper, please add some discussions to state the technical contributions.
2. The authors claimed such a synthetic way could be useful for rare diseases. Please provide the evidence.

---

> ### Author Rebuttal · Authors · 2023-08-08
>
> We thank the reviewer for the thoughtful comments. Please refer to the combined response above for a general overview of the improvements and changes that have been incorporated in the revised manuscript.
>
> **Weaknesses**
>
> 1- We have prepared a Github repository with all the codes required to reproduce our experiments. The codes will be released upon acceptance and the end of the anonymity period.
>
> 2- As you mentioned, we have indeed demonstrated the relationship between the amount of expert feedback and model performance in **Figure 4 a) and b)**. Our results show that as we incorporate more feedback, the quality of synthetic images improves. Regarding the use of more rules or alternative sets of rules, we agree that this could potentially enhance the model performance. However, due to time constraints, we were unable to explore this aspect in depth. Nevertheless, we have ensured that the rules used in our study are comprehensive and based on established clinical knowledge found in relevant textbooks. These rules were carefully chosen to capture the complex image quality and subtle differences between cell types, which are crucial for generating clinically plausible synthetic medical images. In our revised manuscript, we will provide a more detailed explanation of the process of defining these rules and their relevance to the evaluation of complex image quality and small image differences. We will also discuss the potential benefits of exploring additional or alternative sets of rules in future studies, as well as the limitations imposed by time constraints.
>
> 3- We appreciate your feedback on the downstream performance evaluation and the suggestion to consider more tasks and explore the performance of downstream models trained using both synthetic and real images.
>
> (1)  We believe that we have demonstrated the effectiveness of our approach through three downstream tasks. First, we showed that the synthetic data quality is improved by both clinician evaluation (**Main Table 3**) and computational-based qualitative measurements (**Main Table 4**). Second, we demonstrated that human feedback could improve the synthetic data's ability for cell type classification in an amount-dependent manner (**Main Table 5, Figure 4 a,b**). Lastly, we showed that specific human feedback could drive the conditional generation of new concepts (**Main Figure 4 c**).
>
> However, your suggestion inspired us to perform another downstream task, where we conducted a Turing test-style experiment: asking a pathologist to distinguish synthetic data from real data and evaluating whether the fine-tuned model can confuse the pathologist more. **Rebuttal Table 5** showed that pathology-in-the-loop framework could indeed improve model’s performance in generating high quality clinical images data that could confuse the pathology expert more, representing by the decrease in accuracy.
>
> (2) We appreciate your suggestion to explore the performance of downstream models trained using both synthetic and real images. We conducted the experiment (**Rebuttal Table 4**). As can be seen from the table, leveraging both real and synthetic data together benefits morphology-based cell classification. We will incorporate these findings into the refined manuscript.
>
> **Limitations**
>
> Thanks for your suggestions. In the revised manuscript, we will add a more elaborate discussion on the technical contribution. Regarding the potentially utilization in diagnosing rare diseases: we would like to highlight two key aspects of our pathologist-in-the-loop framework that support its potential application in this context:
>
> 1.	Our framework facilitates the generation of high-quality synthetic images for rare cell types, which may have a frequency of less than 1% in healthy individuals. While common image generation pipelines might struggle to generate high-quality images due to the low frequency of these cell types, our pathologist-in-the-loop framework helps improve image quality. This can be particularly valuable in diagnosing rare diseases where high-quality images of rare cell types are essential.
>
> 2.	Our method, as described in **Subsection 2.4**, allows for the rapid incorporation of new concepts into trained medical image generative models. The concept introduced in this study for differentiating subtypes based on nuclear morphology can potentially be extended to identify pathological variations observed in cancer diagnoses, such as changes in nuclear size, shape, texture, and nuclear: cytoplasmic ratio. Additionally, this approach could be applied to cytoplasmic features. For rare diseases, generating synthetic data using our framework may provide the best option for creating sufficient training data to develop and evaluate AI models effectively.

---

> > ### Comment · Reviewer_Mvby · 2023-08-13
> >
> > Overall, I am satisfied with the author's response and hope these comments could improve the final version of this paper.

---

### Official Review · Reviewer_ZdBm · 2023-07-05

**Soundness:** 3 good
**Presentation:** 4 excellent
**Contribution:** 2 fair
**Rating:** 6
**Confidence:** 4

**Summary:**

This paper develops a framework that generates synthetic medical images aligned with the clinical knowledge of doctors through training a reward model based on pathologists' image plausibility annotations. The intuition for their very simple but powerful approach that depends on incorporating clinical expert knowledge into the diffusion models by using the reward model to inform a finetuning objective arises from their observation: "designing domain-specific objective functions for finetuning foundational models that ensure a generative model adheres to clinical knowledge is challenging". Their presentation is very clear, and empirical evaluations & ablations are convincing. I recommend this paper for acceptance beyond my concerns around how well this paper fits into NeurIPS venue.



**Strengths:**

- Very clear presentation
- Simple but powerful idea that brings in an idea from a different domain (LLMs) to medical imaging
- Convincing empirical results and ablations

**Weaknesses:**

My main worry about this paper is that NeurIPS might not be the best venue, given the algorithmic novelty is limited here. However, I do think this is an interesting application of integrating domain-specific knowledge to models through RLHF-style training.

**Questions:**

- How do you think your findings might change as the diffusion models get stronger in creating more plausible images (given the rate of implausible images was higher than 80% for some cell types)?
- How do you think you might generalize your method to other modalities such as interpreting MRI images where clinical plausibility criteria might be more limited? (might be harder to define the criteria beyond aliasing artifacts in certain anatomies?)

**Limitations:**

Authors have not explicitly described limitations of their work.

---

> ### Author Rebuttal · Authors · 2023-08-08
>
> We thank the reviewer for the thoughtful comments. Please refer to the combined response above for a general overview of the improvements and changes that have been incorporated in the revised manuscript.
>
> **Weaknesses**
> This paper serves as an application submission to NeurIPS, aligning with the conference's well-established track for such submissions. While our paper is not motivated by a core technical problem, we believe that the application of ideas from the RLHF to clinical problems is by itself a novel idea that might have significant impact in our application domain of interest. We believe that presenting our work at NeurIPS would provide valuable insights to the broader AI and machine learning community, demonstrating the potential impact of integrating domain-specific knowledge in real-world applications.
>
> **Questions**
>
> 1- As diffusion models become stronger and generate more clinically plausible images, we anticipate that the rate of implausible images would decrease. However, the importance of incorporating human feedback in the loop would still remain significant for several reasons:
>
>     (1)	Ensuring clinical validity: Even with stronger diffusion models, it is crucial to align the generated images with clinical knowledge. Expert feedback would continue to play a vital role in refining the model's performance and ensuring clinical validity for various cell types and rare diseases.
>
>     (2)	Learning new clinical concepts: Human feedback not only helps improve the quality of synthetic images but also teaches the model new clinical concepts not annotated in the original training data. This aspect would remain valuable regardless of the initial strength of the diffusion model.
>
>     (3)	Adapting to evolving medical knowledge: Medical knowledge and best practices continue to evolve over time. Incorporating human feedback allows the model to stay up-to-date with the latest clinical understanding and maintain its relevance and utility in the ever-changing healthcare landscape.
>
>     In conclusion, while we acknowledge that stronger diffusion models may generate more plausible images, the integration of human feedback would still be crucial for ensuring clinical validity, learning new concepts, and adapting to evolving medical knowledge.
>
> 2-  We acknowledge the potential difficulties in defining criteria for modalities like MRI; however, we believe our method can still be adapted and generalized with some modifications:
>
>     (1)	Collaborating with domain experts: For modalities like MRI, it is essential to work closely with radiologists or other relevant domain experts who possess the necessary knowledge to identify and define the clinical plausibility criteria. Their expertise would help to set guidelines and provide feedback on the generated images.
>
>     (2)	Developing modality-specific reward models: To account for the unique characteristics and challenges associated with different imaging modalities, we can develop modality-specific reward models. These models would be trained to predict expert feedback on the generated images for the particular modality, ensuring that the human-in-the-loop framework remains effective in refining the generated images' clinical plausibility.
>
>     (3)	Leveraging auxiliary data: In cases where clinical plausibility criteria are harder to define, we can leverage auxiliary data, such as clinical reports, annotations, or patient metadata, to guide the model training and evaluation. This additional information can help the model learn more nuanced and context-specific criteria for generating clinically plausible images.
>
>     (4)	Incorporating active learning techniques: In situations where defining clinical plausibility criteria is challenging, active learning techniques can be employed to selectively query expert feedback on the most uncertain or ambiguous cases. This approach would ensure that the model gains the most valuable information from experts, even when the criteria are not easily defined.
>
>     In conclusion, we believe that our method can be generalized to other imaging modalities, such as MRI, by closely collaborating with domain experts, developing modality-specific reward models, leveraging auxiliary data, and incorporating active learning techniques. These adaptations would help ensure the effectiveness of our human-in-the-loop framework in generating clinically plausible images across various modalities.

---

> > ### Comment · Reviewer_ZdBm · 2023-08-18
> >
> > I thank the authors for their explanations, my assessment remains unchanged.

---

### Official Review · Reviewer_n6aW · 2023-07-11

**Soundness:** 3 good
**Presentation:** 3 good
**Contribution:** 3 good
**Rating:** 7
**Confidence:** 4

**Summary:**

The authors proposed to include pathologist in the loop for medical data synthesis. The pathologist can be replaced by training a reward model, which is used to fine-tune the generation model.

**Strengths:**

- I admire that the authors broke down the barriers between disciplines, and I believe their proposed pathologist-in-the-loop synthetic data generation framework is a promising strategy for improving the generation of biomedical samples. It is the main reason why I give a positive score. As "pathologist-in-the-loop" is the key contribution, I personally encourage that the raw scores from doctors and the corresponding images should be published in pair after acceptance -- then we can check the quality of the manual annotations.

- The paper is well-organized and the experiments are comprehensive.

**Weaknesses:**

- The authors only gives the overall performances. I wonder if there is a direct verification for the reward model.
- It would be better if this idea can be evaluated on other datasets. Of course, I know it is hard.
- The technical innovation seems to be limited.

**Questions:**

A direct evaluation of the reward model is helpful. I am curious about the relationship of the reward model quality and the synthesis performances.

**Limitations:**

The authors claimed that they have discussed the limitations, but it is not clear for me. Maybe I missed something.

---

> ### Author Rebuttal · Authors · 2023-08-08
>
> We thank the reviewer for the thoughtful comments. Please refer to the combined response above for a general overview of the improvements and changes that have been incorporated in the revised manuscript.
>
> **Weaknesses**
>
> 1- Thanks for suggesting a direct verification of the reward model. We provide the reward function performance on a held-out validation set, which is also annotated by clinical experts (**Rebuttal Table 2**, Please refer to the pdf attached with the combined response to check the results). The reward function performance improves with more feedback, which correlates with the downstream classification performance demonstrated in **Main Figure 4**. In the revised manuscript, we will include detailed performance metrics for each cell type in the appendix, rather than showing only the overall performance. This should provide a more comprehensive understanding of the model's effectiveness.
>
> 2- We concur that assessing our methodology on additional datasets is crucial to validate its applicability across various contexts. To address this concern, we have attempted to gather another independent dataset from an external institute. Regrettably, due to time constraints, we were only able to annotate 20 cell types per class for the training set, a significantly smaller number compared to the 128 per class in the dataset presented in the paper. Despite this limitation, we proceeded to train a diffusion model as the baseline and employed the same workflow to fine-tune it. The reward function remained consistent; however, it was applied to synthetic images generated from the new and independent dataset. The results are shown in the **Rebuttal Table 3** (Please refer to the pdf attached with the combined response to check the results).
>
> Owing to the limited data, the baseline performance was substantially inferior to that of the primary dataset. Nonetheless, the pathologist-in-the-loop approach still managed to significantly enhance the model's performance. This indicates that our method is effective even when applied to smaller and distinct datasets.
>
> 3- We believe that the novelty and significance of our work lie in its application and the unique combination of existing techniques to address the challenges of generating clinically plausible synthetic medical images. To the best of our knowledge, our study is the first to leverage human feedback to model clinician preferences in the context of medical image generation.

---

> > ### Comment · Reviewer_n6aW · 2023-08-12
> > **After Rebuttal**
> >
> > Thank you for you reply, and the responses with revised paper solve my concerns. But, I still hope the author can refine the work mentioned in the second point and present the final results in the final version. Thank you!

---

### Official Review · Reviewer_q99y · 2023-07-20

**Soundness:** 4 excellent
**Presentation:** 4 excellent
**Contribution:** 4 excellent
**Rating:** 8
**Confidence:** 5

**Summary:**

This paper introduces a pathologist-in-the-loop framework for generating clinically plausible synthetic medical images. The training process is similar to generative adversarial nets, with two major modifications. First, the discriminator was trained by human input instead of real/fake labels. Second, the generator was replaced by Diffusion Models. Evaluation of synthetic bone marrow patches by expert hematopathologists, leveraging thousands of feedback points, demonstrates the significant quality enhancement achieved through human input.

**Strengths:**

1. I found the paper to be well-written and informative, and I thoroughly enjoyed reading it.

2. The generation of synthetic data holds significant importance.

3. The extensive visualization of the synthetic images, along with the notable improvement resulting from the pathologist's involvement, is highly convincing.

**Weaknesses:**

1. Missing an important baseline method that incorporates feedback from a real/fake binary classifier. This classifier can distinguish between plausible (real) and implausible (fake) images without relying on pathologists' feedback or cell-type labels (automated feedback). It is imperative to investigate whether utilizing this information can enhance the ability of Diffusion Models to generate realistic images.

2. The clinical plausibility criteria presented in Table 1 pose challenges for implementation in various other tasks. Firstly, developing the checklist items necessitates domain expertise. Secondly, disagreements among pathologists may arise regarding these criteria.

**Questions:**

1. Despite incorporating 100% pathologist feedback, there appears to be a discernible disparity between synthetic and real data. Could you elaborate on the underlying reasons for this gap and propose potential solutions to mitigate it?

2. What if pathologists are unable to differentiate between real and synthetic images? For example, in some cases [1], the experts may not be able to distinguish the real/synthetic images.

3. What is the rate of implausible images after fine-tuning the diffusion model?

4. Consider integrating both real and synthetic data for training purposes. To what extent can this integration enhance the overall performance? This is particularly important as the authors have not yet achieved comparable performance of AI models trained solely on synthetic data when compared to those trained on real data.

**Reference**

[1] Hu, Qixin, Yixiong Chen, Junfei Xiao, Shuwen Sun, Jieneng Chen, Alan L. Yuille, and Zongwei Zhou. "Label-free liver tumor segmentation." In Proceedings of the IEEE/CVF Conference on Computer Vision and Pattern Recognition, pp. 7422-7432. 2023.

**Limitations:**

No potential negative societal impact was detected.

---

> ### Author Rebuttal · Authors · 2023-08-08
>
> We thank the reviewer for the thoughtful comments. Please refer to the combined response above for a general overview of the improvements and changes that have been incorporated in the revised manuscript.
>
> **Weaknesses**
>
> 1- We appreciate the reviewer's suggestion to include a baseline method that incorporates feedback from a real/fake binary classifier. This classifier would distinguish between plausible (real) and implausible (fake) images without relying on pathologists' feedback or cell-type labels, offering automated feedback. Following your suggestion, we conducted experiments where we trained a binary classifier, referred to as a naive classifier, on real and artificial images without relying on pathologists' feedback or cell-type labels. However, this naive classifier did not improve the diffusion model's performance in the downstream cell-typing task, as shown in **Rebuttal Table 1** (Please refer to the pdf attached with the combined response to check the results).
>
> In summary, while the real/fake binary classifier offers an interesting approach, our experiments show that it does not lead to improvements in the downstream cell-typing task, which highlights the importance of pathologists' feedback to generate synthetic medical images that hold clinical validity.
>
> 2-  We acknowledge that developing these criteria requires domain expertise and that disagreements among pathologists may arise. We would like to address these concerns as follows:
>
>     (1)	Domain expertise: It is indeed true that annotating the synthetic medical image quality requires expert knowledge, which is why we have engaged clinician experts to help with the annotation process. We believe that the expert-informed criteria provide a robust foundation for evaluating the clinical plausibility of generated images. Furthermore, once this annotation process is completed, the criteria can be potentially applied to other tasks within the domain, such as cell counting and abnormal cell identification. This makes our approach scalable and applicable to a wide range of medical imaging tasks.
>
>     (2)	Disagreements among pathologists: While we acknowledge that disagreements among experts may occur, the criteria we have used are based on standards outlined in medical textbooks. We have made every effort to ensure that our criteria are as unbiased and objective as possible.
>
> **Questions**
>
> 1- There are several factors that may contribute to the observed disparity between synthetic and real data:
>
>     (1)	Limitations of the generative model: While our generative model is capable of synthesizing visually realistic images, it may not fully capture the complex and nuanced features present in real medical images. This is an inherent challenge in generative modeling, especially when dealing with high-dimensional and intricate data such as medical images.
>     (2)	Incomplete representation of clinical knowledge: Although we have incorporated pathologist feedback, there might be aspects of clinical knowledge that are not entirely captured by the reward function. Medical expertise is vast, and it can be challenging to encapsulate all relevant information in a single model.
>
> 2- Please note our feedback collection process requires pathologists to identify images that are not clinically valid, and not differentiate between real/synthetic images. In fact, pathologists are only shown synthetic images for feedback. If the pathologist cannot distinguish between real and synthetic images, this means that a synthetic image is clinically valid.
>
> 3- As shown in **Main Table 3**, we found that incorporating pathologist feedback led to a significant boost in the quality of synthetic images across all cell types. After fine-tuning the diffusion model with expert feedback, the average rate of clinical plausibility increased from 0.21 to 0.75. This improvement demonstrates the effectiveness of our approach in generating more clinically plausible synthetic medical images.
>
> 4- Integrating real and synthetic data for training purposes can potentially enhance overall performance; however, it is essential to ensure that the synthetic data is of high quality and clinically plausible. Our current approach aims to improve the quality of synthetic medical images by incorporating pathologist feedback, which can make synthetic data more suitable for integration with real data during the training process. It is important to note that the synthetic data generated in our approach is based on the limited real data available. As a result, integrating synthetic and real data together may not fully address all the issues arising from the limited real data, such as underrepresented classes or rare cases.

---

> > ### Comment · Reviewer_q99y · 2023-08-12
> >
> > I thank the authors for the detailed responses to my concerns and questions. I think it is a great work incorporating domain knowledge from expert humans and generative models, which is naturally expected to be beneficial than simply let the model learn to synthesize from raw data. I hope the authors can include the some discussion in the review phase to the final version.

---

### Official Review · Reviewer_uSrF · 2023-07-27

**Soundness:** 3 good
**Presentation:** 4 excellent
**Contribution:** 3 good
**Rating:** 7
**Confidence:** 4

**Summary:**

This paper proposes a pathologist-in-the-loop framework for generating clinically plausible synthetic medical images using diffusion models. The process involves pretraining a conditional diffusion model on real medical images, then using synthetic images labeled by expert pathologists to train a "clinical plausibility reward model" to predict pathologist feedback on new images. Finally, the diffusion model is fine-tuned using a reward-weighted likelihood loss that incorporates the reward model, to align synthetic outputs with clinical knowledge.
The method is evaluated on a bone marrow cell image dataset. Results show the proposed method incorporating pathologist feedback significantly improves the clinical plausibility, fidelity, diversity, and downstream utility of synthetic images.

**Strengths:**

1) Clinically plausible medical image synthesis is an interesting and important problem. This paper provides a promising human-in-the-loop framework to incorporate clinical knowledge.
2) The method is evaluated extensively both qualitatively and quantitatively. The results demonstrate clear improvements from incorporating expert feedback.
3) The paper is well-written and easy to follow. The method is described in sufficient detail.

**Weaknesses:**

1) The criteria used by the pathologist for judging clinical plausibility could be described more precisely. Are there any quantitative metrics for each criterion?
2) Only binary feedback is collected from the pathologist. More fine-grained ratings could provide richer supervision signal.
3) Evaluations could be conducted on multiple datasets and with higher resolution images.
4) The comparison to "automatic feedback" using one classifier is somewhat weak. Better baselines or more comparisons could be evaluated.

**Questions:**

1) The proposed method is only evaluated on a single dataset. Experiments on more datasets may be necessary to validate the generalizability of the method. Experimenting on higher resolution image synthesis could also be important for potential clinical usage.
2) It might be helpful to include more visual or qualitative comparisons between synthetic images marked as "implausible“ or "plausible".
3) Ye et al. also proposed a relevant idea using automatic feedback as the reward in [1]. It might be worth discussing this paper.
[1] Ye et al. Synthetic Sample Selection via Reinforcement Learning. MICCAI 2020.

**Limitations:**

Authors could consider discussing the limitations and potential negative societal impact of this work:

Limitations:
1) The method has only been evaluated on one medical imaging modality/dataset (bone marrow). Applicability to other datasets or modalities with different visual and clinical characteristics is unclear.
2) The framework requires extensive pathologist time for labeling synthetic images. Scalability to large diverse datasets may be problematic.
3) Binary plausibility labels may not capture nuanced aspects of clinical knowledge. More granular ratings could be beneficial.
4) The synthetic images are with low resolution (64x64). Quality and utility for diagnosis may degrade for higher resolution synthesis.

Example negative societal impacts:
1) If not carefully validated, inaccuracies in synthetic medical images could mislead or harm practitioners relying on them for diagnosis/treatment.
2) Widespread availability of synthetic medical data could lead to privacy risks if reconstructed from patient data without consent.

---

> ### Author Rebuttal · Authors · 2023-08-08
>
> We thank the reviewer for the thoughtful comments. Please refer to the combined response above for a general overview of the improvements and changes that have been incorporated in the manuscript to address this issue.
>
> **Weaknesses**
>
> *1- Criteria for evaluating clinical plausibility*
>
> In the final manuscript, we will further elaborate on the qualitative criteria for evaluating clinical plausibility. However, **we would like to stress that quantitative metrics are not possible as cells can be implausible for a wide range of unpredictable reasons.** This is precisely the motivation behind our work; if there were a finite number of ways in which images can be invalid and if there were quantitative metrics for evaluating those failures, there would be no need for human feedback.
>
> *2- Limitation to binary feedback*
>
> We appreciate the reviewer's suggestion of collecting more fine-grained ratings from the pathologist to provide a richer supervision signal. Indeed, incorporating a more detailed feedback system could potentially enhance the learning process and improve the model's performance. However, there are some practical considerations that led us to opt for binary feedback. Binary feedback simplifies the annotation process for the pathologist, reducing the cognitive load and minimizing potential inconsistencies in the ratings. In a clinical setting, it is crucial to balance the workload of the pathologist while ensuring that the feedback provided is both accurate and meaningful. Furthermore, our results suggest that binary feedback is an effective supervision signal for generating clinically plausible synthetic medical images. That being said, we acknowledge the potential benefits of more fine-grained ratings and will consider exploring this approach in future work.
>
> *3- Image resolution*
>
> We would like to clarify the context in which these images are generated and used, as it may help address the concern about the resolution.  The resolution of the single cell patch may appear low, but these are cropped from whole slide images scanned at 400x magnification, which is the industry standard for clinical whole slide scanners.
>
> It is important to note that clinicians routinely use this resolution, or even lower resolution, for cell counting and making diagnoses. Thus, the resolution of the synthetic images in our study is consistent with the resolution used in real-world clinical settings and should not impact the utility of our approach for diagnosis. Our methodology is designed to be compatible with the standard industry practices and resolution requirements in the clinical context.
>
> *4- The automatic feedback baseline*
>
> We thank this reviewer for suggesting more baseline experiments. Accordingly, we add one more baseline control that incorporates feedback from a real/fake binary classifier. This classifier would distinguish between plausible (real) and implausible (fake) images without relying on pathologists' feedback or cell-type labels, offering automated feedback. Please see our reply to **Reviewer q99y** for the results.
>
> **Answers to Questions**
>
> 1- We leveraged another independent dataset and showed that the framework is generalizable to the external dataset. Please see our response for **Reviewer n6aW** for the results.
>
> 2- We appreciate the reviewer's suggestion to include more comparisons between synthetic images marked as "implausible" and "plausible." As per your suggestion, we have included a visualization of 128 images, wherein 64 images are considered realistic, and the remaining 64 are deemed implausible. This visualization will provide a clear comparison between plausible and implausible synthetic images (**Rebuttal Figure 1**) and please refer to the pdf attached with the combined response to check the results.
>
> Additionally, we have added an appendix to the paper, which contains 512 images, with 32 images for each cell type. We hope that these additional visualizations and comparisons will effectively address the reviewer's concerns and enhance the clarity of our paper.
>
> 3- We appreciate the opportunity to discuss Ye et al., MICCAI 2020. While Ye et al. propose a reinforcement learning (RL) based method for selecting high-quality synthetic samples to improve the performance of medical image recognition systems, our work focuses on improving the generative model itself by incorporating pathologist feedback. We believe that our approach differs from Ye et al.'s work in three significant ways: (1) generative model improvement, (2) task complexity, and (3) expert validation. We will include a comparison in the revised manuscript.
>
> **Limitations:**
>
> 1- Evaluation on a single medical imaging modality/dataset: In the revised manuscript, we will discuss this limitation and emphasize the need for future studies to evaluate our approach on various medical imaging modalities and datasets to ensure its generalizability.
>
> 2- Pathologist time requirements: We will address this limitation in our paper and explore possible ways to alleviate this issue, such as involving multiple experts, leveraging active learning techniques to minimize the number of labels required, or incorporating semi-supervised or unsupervised learning methods.
>
> 3- Binary plausibility labels: In the revised manuscript, we will discuss this limitation and consider potential solutions, such as adopting multi-level or continuous rating scales for clinical plausibility, which may provide a more detailed representation of expert knowledge.
>
> Regarding the potential negative societal impact, we understand that the misuse or biased generation of synthetic medical images could have adverse consequences in medical research and decision-making. In the revised manuscript, we will discuss the importance of ensuring the ethical use of synthetic medical images, addressing potential biases in the generative models, and considering privacy concerns when generating and sharing synthetic data.

---

> > ### Comment · Reviewer_uSrF · 2023-08-17
> >
> > I appreciate authors' detailed rebuttal and additional results/analysis. I enjoyed reading the paper and rebuttal. Since authors have resolved most of my concerns in the rebuttal, I tend to keep my original rating as Accept.

---

### Author Rebuttal · Authors · 2023-08-08

We thank all reviewers for the comprehensive and constructive feedback on our submission. The valuable input received has significantly contributed to improving our work. We have prepared an extensive, point-by-point response for each reviewer, outlining our plans to address their concerns and suggestions for additional analyses to enhance the manuscript. **All new figures and tables can be found in the attached one-page PDF file.** We believe that our response will address the reviewers' concerns and allow us to promptly resolve any remaining minor issues.

**Summary of improvements based on reviewers' comments**

1. External validation:

We agree that evaluating our methodology on additional datasets is essential for validating its applicability across various contexts. To address this, we collected an independent dataset from a separate hospital, and our pathologist-in-the-loop approach still demonstrated significant improvements in the model's performance, indicating its effectiveness across multiple datasets (**Rebuttal Table 3**).

2. More comprehensive comparison and evaluation

We introduced an additional baseline control using feedback from a real/fake binary classifier to distinguish between plausible and implausible images without relying on pathologist feedback or cell-type labels  (**Rebuttal Table 1**). Furthermore, we included a comparison where we trained the downstream classifier with both synthetic and real images (**Rebuttal Table 4**). Lastly, we added a Turing test-style experiment, asking a pathologist to differentiate synthetic data from real data (**Rebuttal Table 5**). These additional experiments collectively enhance the quality and rigor of our work.

3. Validation of the reward model

We presented the reward function performance on a held-out validation set annotated by clinical experts (**Rebuttal Table 2**). The results show that the reward function's performance improves with more feedback, correlating with the downstream classification performance demonstrated in **main Figure 4**. This offers a more comprehensive understanding of the model's effectiveness.

4. A more comprehensive comparison illustrating clinician feedback

To better understand the distinctions between plausible and implausible images, we included a visualization of 128 images (64 plausible and 64 implausible) in **Rebuttal Figure 1**. Additionally, we added an appendix containing 512 images, with 32 images for each cell type, to provide a deeper understanding of the different cell types and variations in image quality. These additional visualizations and comparisons effectively address the reviewers' concerns and enhance the clarity of our paper.

Once again, we would like to thank all reviewers for their comments and we are looking forward to the discussion period.

---

### Decision · Program_Chairs · 2023-09-21

**Decision:**

Accept (spotlight)

**Comment:**

All reviewers land on the positive side. They all enjoy the proposed pathologist-in-the-loop framework. This framework was evaluated extensively both qualitatively and quantitatively, showing its ability to generate clinically-plausible synthetic medical images. The AC suggests acceptance and encourages the authors to include the discussion in the rebuttal into the final version.